# Comparative effectiveness of cervical vs thoracic spinal-thrust manipulation for care of cervicogenic headache: A randomized controlled trial

Gopal Nambi[1]*, Mshari Alghadier[1], Mudathir Mohamedahmed Eltayeb[2], Osama R. Aldhafian[3], Ayman K. Saleh[3,4], Nesreen Alsanousi[5], Alaa Jameel A. Albarakati[6], Mohamed A. Omar[4], Mohamed Nagah Ahmed Ibrahim[4], Abdehamid A. Attallah[4], Mohammed Abdelgwad Ismail[4], Mohamed Elfeshawy[4]

1 Department of Health and Rehabilitation Sciences, College of Applied Medical Sciences, Prince Sattam Bin Abdulaziz University, Al-Kharj, Saudi Arabia, 2 Department of Nursing, College of Applied Medical Sciences, University, Prince Sattam Bin Abdulaziz University, Al-Kharj, Saudi Arabia, 3 Department of Surgery, College of Medicine, Prince Sattam Bin Abdulaziz University, Al-Kharj, Saudi Arabia, 4 Faculty of Medicine for Girls, Department of Orthopedic Surgery, Al-Azhar University, Cairo, Egypt, 5 Department of Biochemistry, College of Medicine, Prince Sattam Bin Abdulaziz University, Al-Kharj, Saudi Arabia, 6 Department of Surgery, College of Medicine, Umm Al-Qura University, Al-Qunfudah Branch, Makkah, Saudi Arabia

* physio_gopal@rediffmail.com

**Data Availability Statement:** The minimal data set is available as a supporting file.

## Abstract

### Background

There is ample evidence supporting the use of different manipulative therapy techniques for Cervicogenic Headache (CgH). However, no technique can be singled as the best available treatment for patients with CgH. Therefore, the objective of the study is to find and compare the clinical effects of cervical spine over thoracic spine manipulation and conventional physiotherapy in patients with CgH.

### Design, setting, and participants

It is a prospective, randomized controlled study conducted between July 2020 and January 2023 at the University hospital. N = 96 eligible patients with CgH were selected based on selection criteria and they were divided into cervical spine manipulation (CSM; n = 32), thoracic spine manipulation (TSM; n = 32) and conventional physiotherapy (CPT; n = 32) groups, and received the respective treatment for four weeks. Primary (CgH frequency) and secondary CgH pain intensity, CgH disability, neck pain frequency, neck pain intensity, neck pain threshold, cervical flexion rotation test (CFRT), neck disability index (NDI) and quality of life (QoL) scores were measured. The effects of treatment at various intervals were analyzed using a 3 × 4 linear mixed model analysis (LMM), with treatment group (cervical spine manipulation, thoracic spine manipulation, and conventional physiotherapy) and time intervals (baseline, 4 weeks, 8 weeks, and 6 months), and the statistical significance level was set at $P < 0.05$.

**Funding:** This study is supported via funding from Prince Sattam bin Abdulaziz University project number (PSAU/2023/R/1444). There was no additional external funding received for this study. The funders had no role in study design, data collection and analysis, decision to publish, or preparation of the manuscript.

**Competing interests:** The authors have declared that no competing interests exist.

**Abbreviations:** CgH, Cervicogenic Headache; ROM, range of motion; FRT, flexion rotation test; ICHD, International Classification of Headache Disorders; CSM, cervical spine manipulation; TSM, thoracic spine manipulation; CMT, conventional massage therapy; HVLAT, high-velocity low amplitude thrust; NDI, Neck disability index; QOL, Quality of life.

# Results

The reports of the CSM, TSM and CPT groups were compared between the groups. Four weeks following treatment CSM group showed more significant changes in primary (CgH frequency) and secondary (CgH pain intensity, CgH disability, neck pain frequency, pain intensity, pain threshold, CFRT, NDI and QoL) than the TSM and CPT groups (p = 0.001). The same gradual improvement was seen in the CSM group when compared to TSM and CPT groups (p = 0.001) in the above variables at 8 weeks and 6 months follow-up.

# Conclusion

The reports of the current randomized clinical study found that CSM resulted in significantly better improvements in pain parameters (intensity, frequency and threshold) functional disability and quality of life in patients with CgH than thoracic spine manipulation and conventional physiotherapy.

# Trial registration

**Clinical trial registration:** CTRI/2020/06/026092 trial was registered prospectively on 24/06/2020.

# Introduction

Globally, headache disorders affect approximately 66% of the population between the ages of 18 and 65 years at least once a year. Sixty-six percent of men and fifty-seven percent of women report headaches at least once in their lifetime which reduces the quality of life, work productivity and increased costs to society [1]. Cervicogenic headache (CgH) is a distinct form of headache and accounts for 17.8% of all headaches, the prevalence rate is between 15 and 20% among cases of chronic headache [2]. The prevalence rate of CgH is 0.21% in females and 0.13% in males and has various causative factors [3]. It has a significant negative socioeconomic impact and is a burden on the community and public health [4]. The causative factor for the headache is located in the neck region and the pain is made worse by movements of the head and neck [2]. The most accepted mechanism of CgH is found between the trigeminal nerve and C1 –C3 nerves in the trigemino-cervical nucleus [5]. It usually arises from musculoskeletal structures such as the cervical vertebra, intervertebral disc, or paravertebral muscles. The clinical features of CgH include unilateral or bilateral headache, limited range of motion (ROM) of the neck, and radiating pain to the head or face region [6].

Generally, CgH is diagnosed based on a detailed history and clinical assessment [7]. Physical examinations typically reveal pain in the cervical region–neck pain (NP), decreased neck movements, upper quarter muscle tightness and loss of muscle properties such as excitability, contractility, extensibility and elasticity [8]. The cervical flexion rotation test (CFRT) is a valid, reliable and accurate method with an overall diagnostic accuracy of 91% for assessing cervical range of motion in patients with CgH [9]. The management of CgH consists of pharmacological and non-pharmacological methods, in which the pharmacological means are associated with many side effects such as damage to the liver or kidneys, diarrhea, constipation and allergic reactions [10]. There are also many non-pharmacological treatment modalities available such as; physical modalities, positional release therapy (PRT), muscle strengthening exercises,

ergonomic guidance and patient education etc [11]. It has been estimated that 34% of US citizens receive some sort of physiotherapy for CgH each year [12].

In physiotherapy, joint mobilization and manipulation are the most commonly used treatment modalities for treating patients with CgH [13]. The manipulation technique commonly used to treat CgH targets two different regions in the spine such as the cervical and thoracic spine. Cervical spine manipulation (CSM) and thoracic spine manipulation (TSM) technique uses high velocity, low amplitude thrusts (HVLAT) manoeuvre. Some studies have looked solely at the effects of manipulating the cervical spine in cases of Cervicogenic headache [6,14,15]. Dunning JR et al investigated that six to eight sessions of upper cervical and upper thoracic manipulation were shown to be more effective than mobilization and exercise in patients with CgH, and the effects were maintained at 3 months [16]. Haas et al. investigated the effectiveness of cervical manipulation in patients with CgH [17]. Similarly, McDevitt AW et al. found that thoracic spine manipulation alone significantly improved neck-related disability in CgH, but had no effect on headache-related disability but participants reported overall improvement in their condition [18]. However, so far no studies have compared and investigated the individual effects of cervical spine manipulation, thoracic spine manipulation or conventional physiotherapy in treating patients with CgH.

Numerous studies have supported the application of various manipulative therapy approaches for the treatment of CgH [13–18]. Nevertheless, evidence is scarce in comparing the individual effects of cervical and thoracic manipulation approaches in Cervicogenic headache, particularly regarding its clinical and functional aspects. Additionally, no studies have attempted to address the shortcomings and gaps observed in the existing literature on the management of CgH, such as a lack of comparison between manipulation in two different regions, poor study designs, quality and small sample sizes. Therefore, our study objective was to compare and investigate the individual effects of cervical and thoracic manipulation techniques on patients with CgH. This randomized clinical trial hypothesized that there would be differences in primary and secondary outcome measures between cervical spine manipulation, thoracic spine manipulation, and conventional physiotherapy for treating patients with CgH.

## Materials and methods

### Study design

The trial was a parallel-group, prospective, randomized controlled trial. The required participants were screened and diagnosed by a physician at the University hospital between 1st July 2020 and 31st July 2022 following the CgH diagnostic criteria 11.2.1 from the ICHD-3 (International Classification of Headache Disorders) [19] and the disease CgH falls under the International classification of disease -10 (ICD-10) code of G44. 841 [7]. Ninety-six (N = 96) participants who fulfilled the eligibility criteria were randomly allocated into three groups equally: the cervical spine manipulation (CSM; n = 32), thoracic spine manipulation (TSM; n = 32), and conventional physiotherapy (CPT; n = 32) groups through a computer-generated simple random table and the allocation of the participants to each group was concealed using sealed envelopes. The computer did not generate the group until it was time to randomize each participant, ensuring that the allocation was concealed. No significant changes were made while the study was being carried out because it was designed as a follow-up to a pilot study and the 6-month follow-up data collection was completed on 31st January 2023.

The research was conducted at Physiotherapy OPD, Prince Sattam bin Abdulaziz University, Al Kharj, Saudi Arabia, and the Department Ethical Committee (DEC) granted ethical approval under the reference number RHPT/019/042. The DEC accepted the study protocol as well as the informed consent forms. The study involved human participants who followed the

instructions outlined in the Declaration of Helsinki (1964) and were prospectively registered in clinical trial.gov.in CTRI/2020/06/026092 on June 24, 2020.

## Participants

Participants aged between 18–60 years and suffering from unilateral or bilateral CgH (>3 months) were allowed to participate in the study. Patients with pain intensity ≥3 on a visual analog scale (VAS), CgH resulting from pain in the neck followed by headache, limited neck movements, neck stiffness and cervical spine disorders were allowed to participate in the study. Other primary headaches such as migraine and tension-type headaches (TTH), whip-lash injuries, participants who showed signs of the five 'D's (dizziness, drop attacks, dysarthria, dysphagia, diplopia) or who had signs of the three 'N's (nystagmus, nausea, other neurological symptoms (cord compression or nerve root involvement), contraindications to manipulative therapy (tumour, degenerative and inflammatory arthritis, osteoporosis, dislocation, fractures, and steroid intake), underwent previous head and neck surgeries, had physiotherapy or other complementary therapies in the last three months were excluded. The flow of the study program was documented by following the CONSORT guidelines and is displayed in (Fig 1) [20].

## Interventions

Certified physiotherapists having 10–15 years of experience in providing spinal manipulation for patients with CgH provided the treatment to all the groups. All the participants in the three groups had given their informed consent to participate in the study after understanding the detailed information about the study protocol. All the participants in the three groups received 10 minutes of hydrocollator pack application to relax the muscles of the area around the neck and upper back. Following this, the participant's neck muscles and joints were assessed for any musculoskeletal dysfunction. After that, the participants were given the manipulation techniques as per the directions provided in the study protocol. Standardized treatment techniques were used for all the group participants to reduce intervention bias. The procedures of inter-vention and follow-up measurements were recorded in standardized forms. During the study period, the participants were asked to refrain from taking any other type of intervention, they received the concerned interventions 3 times per week for 4 weeks.

**Spinal manipulation therapy.**   Peterson and Bergman defined spinal manipulation ther-apy (SMT) as a high-velocity low-amplitude thrust (HVLAT) technique [21]. Four experienced physiotherapists having experience in SMT performed this technique after evaluation of each participant by physical examination and palpation. Then the therapist manipulated the upper cervical (C1-2) spine for the CSM group and the upper thoracic (T1-T2) spine for the TSM group by following the study protocol irrespective of joint dysfunction. For both the upper cer-vical (C1-2) and upper thoracic (T1-T2) manipulations, if no popping or cracking sound was heard on the first attempt, the therapist repositioned the patient and performed a second manipulation. A maximum of 2 attempts were performed on each patient. The participants were instructed that the manipulations were likely to be accompanied by multiple audible pop-ping sounds [16]. Because each participant has to receive spinal thrust joint manipulation to their upper cervical or thoracic spines 3 times per week for 4 weeks (12 sessions total), if any participant demonstrated any new red flag signs or showed no signs for manipulation, such as no pain or musculoskeletal dysfunction, then the procedure was not performed.

*Cervical spine manipulation (CSM).* The patient was asked to lie comfortably in a supine position, in which the manipulation targeted the C1–C2 vertebra. The patient's head was kept in a "cradle hold" method" where the head was free from the treatment table. The left posterior arch of the C1 vertebra (atlas) was held with the lateral aspect of the proximal phalanx of the

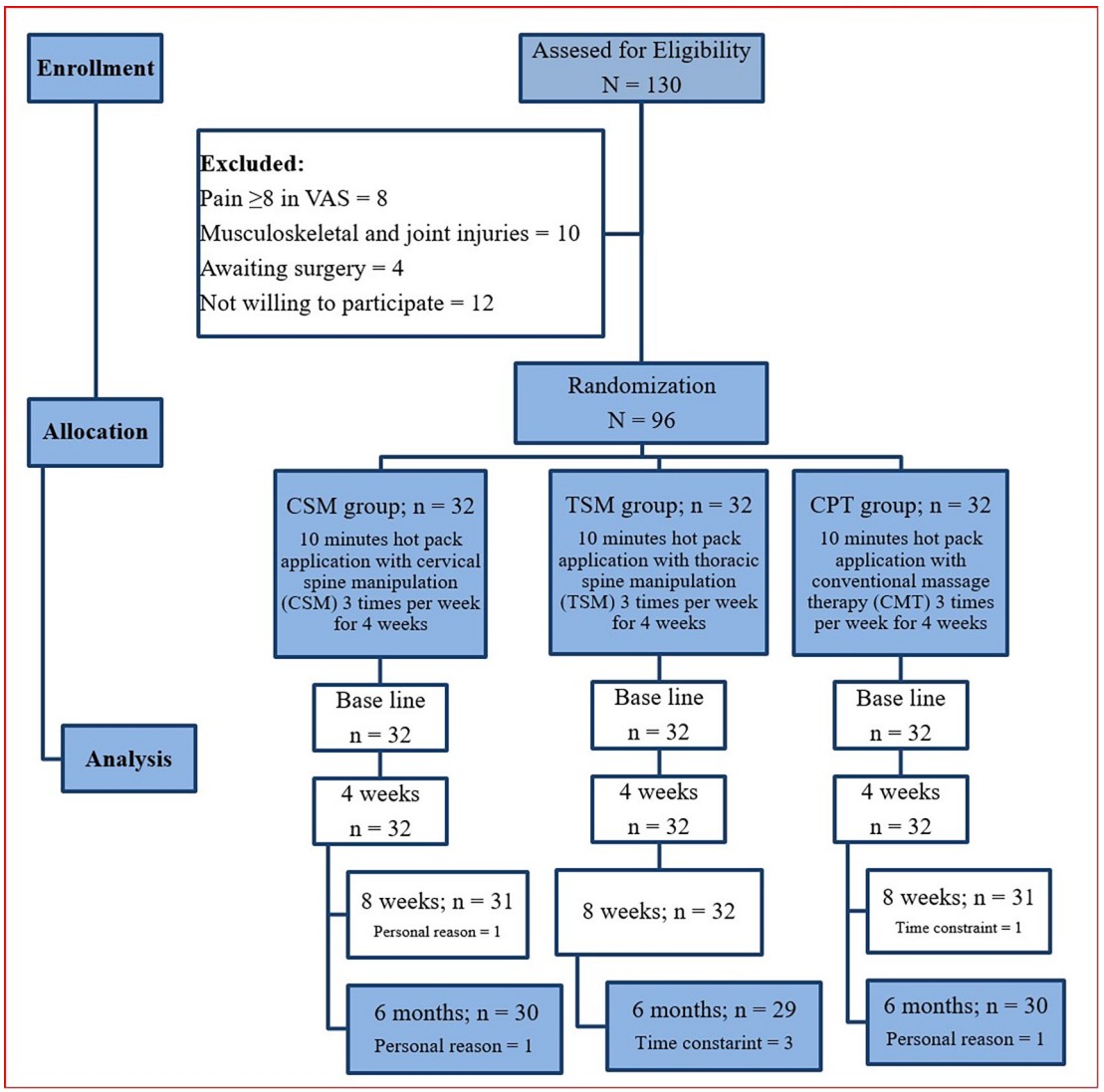

**Fig 1. Flow chart showing the study details.**

therapist's index finger of the left hand. The right hand of the therapist holds the chin of the patient. To localize the forces to the left C1-C2 vertebral articulation, the patient was positioned using an extension, a posterior-anterior (PA) shift, an ipsilateral side-bend and a contralateral side-shift. While maintaining this position, the therapist performed a single high-velocity, low-amplitude thrust (HVLAT) manipulation to the left atlantoaxial joint using right rotation in an arc toward the underside eye and translation toward the table. This was repeated using the same procedure but directed to the right C1-C2 articulation. The selection of the spinal segments to target was prescriptive (C1–C2) and it was based on the study protocol (Fig 2). The manipulation was done first on the pain-free side and then on the painful side and the rotation range was limited by the target vertebra [6].

*Thoracic spine manipulation (TSM).* The patient was asked to lie comfortably in a supine position and the manipulation targeted the T1–T2 vertebra. For this technique, the patient held her/his arms and forearms across the chest with the elbows aligned in a supero inferior direction. The therapist contacted the transverse processes of the lower vertebrae of the target

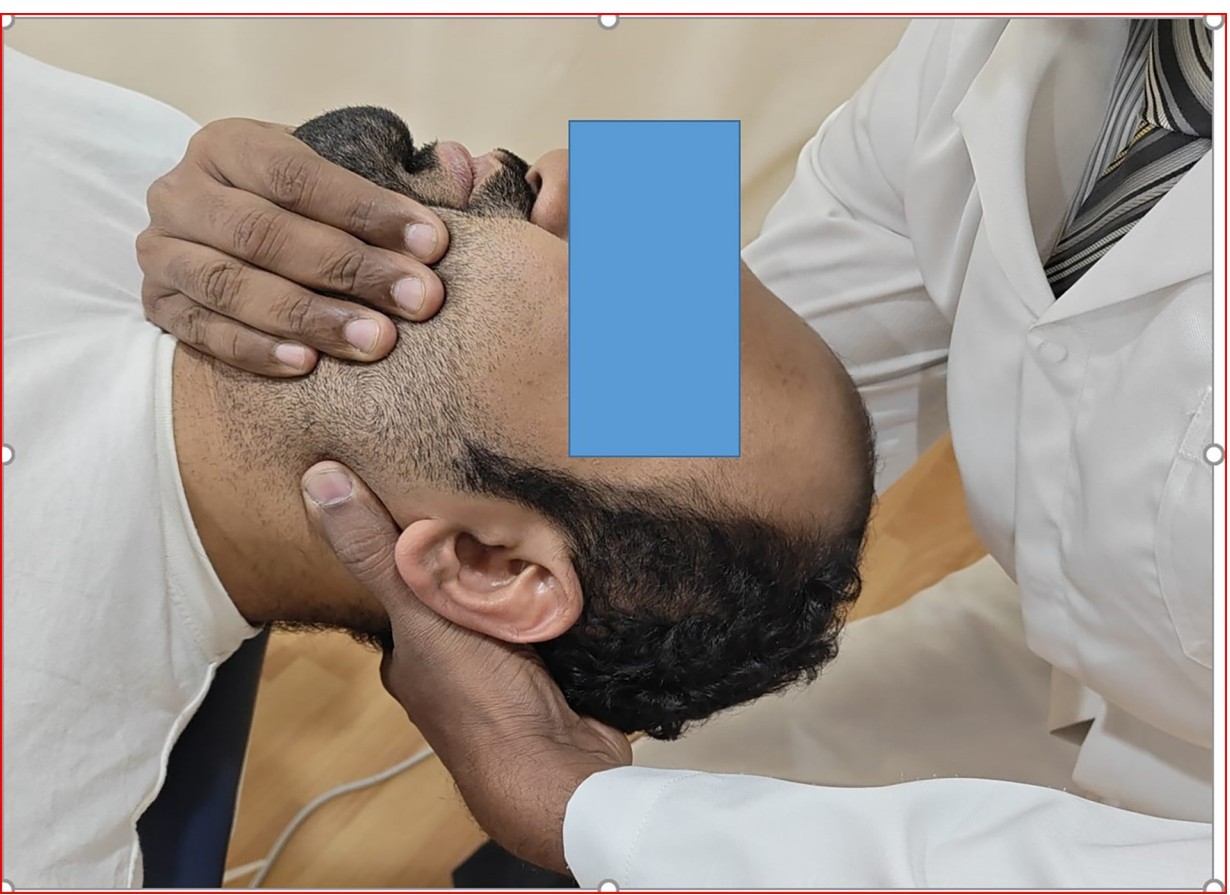

**Fig 2. Figure showing the cervical spine manipulation.**

motion segment with the thenar eminence and middle phalanx of the third digit. The upper lever was localized to the target motion segment by adding rotation away and side-bend towards the therapist while the underside hand used pronation and radial deviation to achieve rotation toward and side-bend away moments, respectively. The space inferior to the xiphoid process and costochondral margin of the therapist was used as the contact point against the patient's elbows to deliver a manipulation in an anterior-to-posterior direction targeting T1-2 bilaterally (Fig 3) [18].

**Conventional physiotherapy (CPT).** The participants of the CPT group received massage therapy for 15 minutes using Queen Helene, Cocoa Butter Face & Body Cream, New York, USA. The participant was asked to lie down in a prone position, with the anterior aspect of the head resting on a face hole in the couch. The treating therapist stands by the patient's head side and uses the tips of the middle fingers of both hands to perform circular kneading on both sides of the C1 to C7 vertebra. This manoeuvre was repeated 3 times for each cervical vertebra, beginning from the C7 vertebra and working towards the C1 vertebra. Then the head was turned to the right side, and the circular kneading was performed on the sub-occipital and paravertebral muscles and the same procedure was done in the left side of the neck [22].

Participants in all three groups received ten minutes of hydrocollator heat before intervention (manipulation or massage). Also, they were asked to perform neck isometric exercises three times a day, every day for 4 weeks. The patient was asked to keep his hand over his forehead and resist the forward movement of his neck for 10 seconds and the same movement was

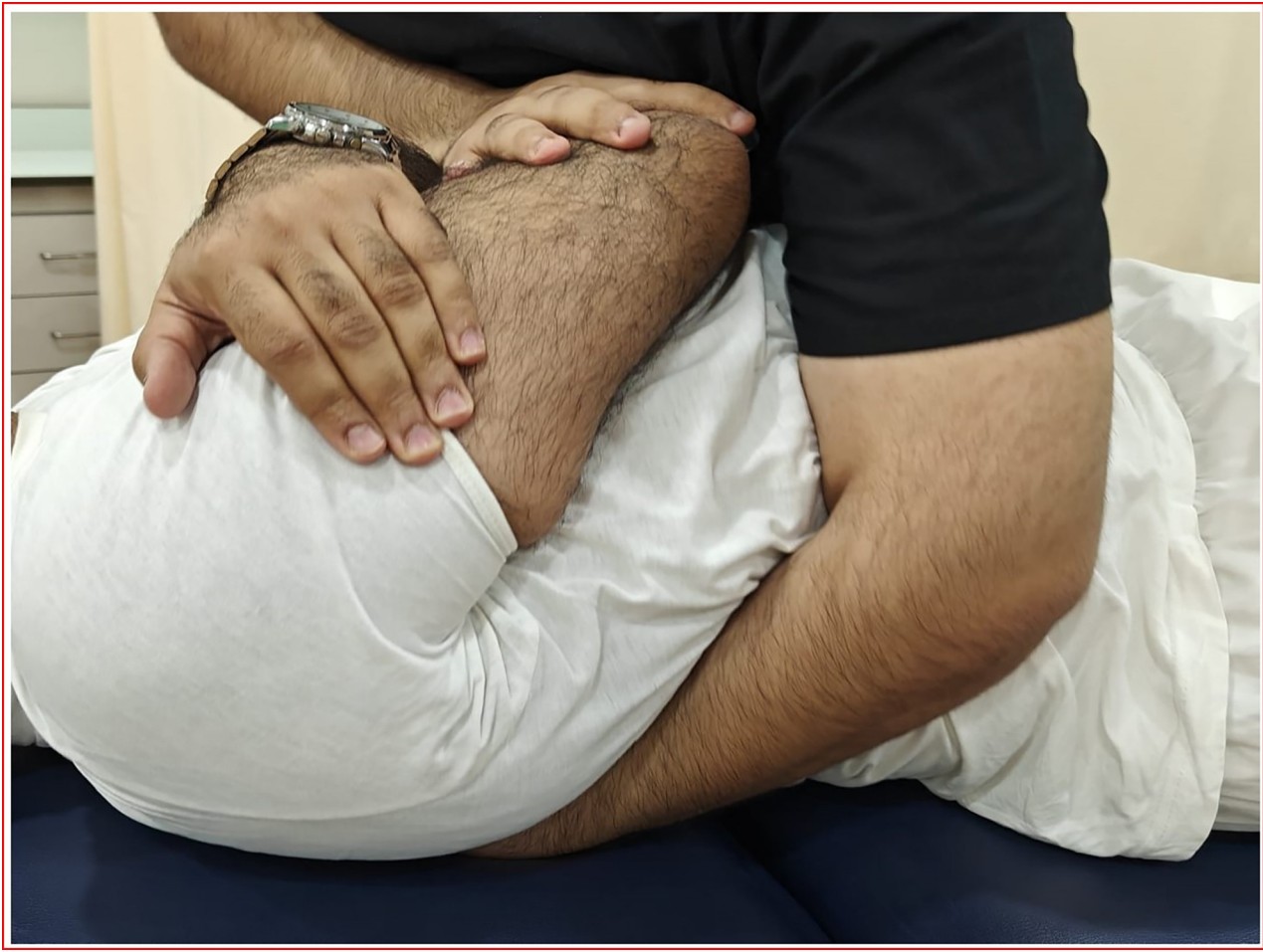

**Fig 3. Figure showing the thoracic spine manipulation.**

repeated 15 times. Similarly, the patient was asked to keep the hand on the posterior and lateral sides of the head and resist the backward and sideways movements of the neck (Fig 4). Also, static active stretching exercises for the upper trapezius, levator scapulae, scalene, and sterno-cleidomastoid muscles were taught to the patients, which was maintained for the 30s with 3 repetitions [23]. The patients were instructed to keep doing this set of exercises after 4 weeks of various intervention protocols and they were asked to maintain an exercise log book to check the treatment compliance.

## Outcomes

All the outcome measures were recorded by a physiotherapist blinded to treatment allocation, and the scores were entered in a data sheet. The scores were measured at the beginning of the study, after 4 weeks, 8 weeks, and at 6 months.

**Primary outcome.** *CgH frequency*: It is a self-administered outcome variable where the patient enters their CgH pain experience in a medical log book every evening to find the number of painful days in 4 weeks [24].

**Secondary outcome.** *CgH pain intensity*: The pain intensity of CgH was assessed using a visual analogue scale (VAS). Patients rated their typical level of pain status during the previous

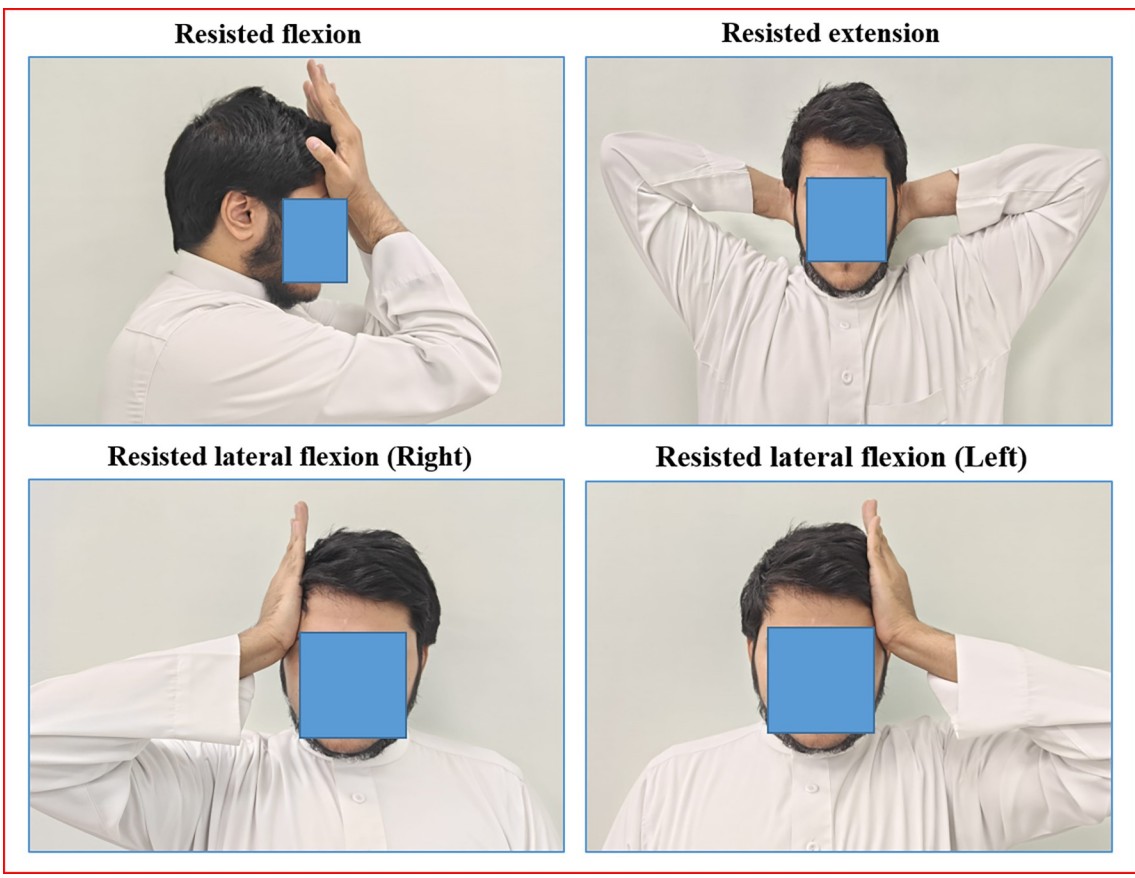

**Fig 4. Figure showing the neck isometric exercises.**

week on a 10 cm horizontal line, with one end 0 representing "no pain" and the other end 10 representing "worst pain imaginable [25]."

**CgH disability**: The Headache Impact Test (HIT) questionnaire is a valid and reliable tool to assess the level of disability in CgH patients. It consists of six items: pain, social functioning, role functioning, vitality, cognitive functioning, and psychological distress. Each item is rated using 5 response categories (never, rarely, sometimes, very often, or always), each category of which is associated with a numerical value (6, 8, 10, 11, and 13, respectively), resulting in a range of possible total summed scores of 36–78. The score categories are no or mild disability (49 or less), moderate disability (50–55), severe disability (56–59), and complete disability (60–78) [26].

**NP frequency**: It is a self-administered outcome variable where the patient enters his neck pain experience in a medical log book every evening to find the number of painful days in 4 weeks [24].

**NP intensity**: The pain intensity of neck pain was assessed using a visual analogue scale (VAS). Patients rated their average pain intensity over the past week on a 10 cm horizontal line, with one end 0 representing "no pain" and the other end 10 representing "worst pain imaginable [25]."

**NP pressure threshold**: It is the lowest intensity at which a given stimulus is perceived as painful and it was measured using an instrument called an Algometer (Baseline, 22-pound dolorimeter, ID, USA). The tip was placed over the points (trigger point on the upper trapezius muscle) on the neck region which is identified through palpation techniques for each

participant and was marked in the skin for further measurement by the blinded therapist. It is a reliable and valid tool for determining pain threshold [27].

***Cervical flexion–rotation test (CFRT):*** The cervical flexion–rotation test is done with the patient in a supine lying position. The therapist passively maintains the patient's neck into full flexion to relax the structures of the middle and lower cervical spine, and then the patient's head is passively rotated in each direction while the flexed position is maintained and the range of motion is measured [9].

***Neck disability index (NDI):*** It is a reliable and valid self-reported questionnaire with ten items scored on a 0 to 5 scale. The grades of disability are determined based on the scores obtained, which are as follows: 10–29% mild; 30–49% moderate; 50–69% severe; 70% or more is a complete disability [28].

***Quality of life:*** The EQ-5D (Euro Qol 5D) is a self-administered health-related quality of life (HRQOL) questionnaire, which measures the five dimensions of quality of life. It includes mobility, self-care, usual activities, pain/discomfort, and anxiety/depression. It is used to assess the CgH patients' overall quality of life with a scale of 0 (worst) to 100 (best) [29].

## Sample size

For calculating the number of subjects to be included in the study, the primary outcome measure CgH frequency in days was taken into consideration based on a previous pilot study which found the effect of spinal manipulation in the treatment of CgH, with 10 subjects in each group. Using the G-Power software (version 3.1.9.2; Franz Faul, University of Kiel, Germany), assuming a two-sided $\alpha = 0.05$, and power ($1-\beta = 0.80$), to detect an effect size of 1.2 CgH days with MCID of 4 CgH days and a standard deviation of 0.5 between the groups, approximately 28 samples were required. In assuming a 10% dropout, we enrolled 32 subjects in each group.

## Blinding

Because of the experimental nature of the study methodology, it was not feasible to blind the treating therapist as well as the participants of the study. The therapists who assessed the outcome variables at baseline, 4-weeks, 8-weeks, and 6-months were blinded. Therefore, the therapist providing the treatment and the therapist measuring the data were different individuals. In addition, the outcome-measuring therapist continued to be masked to the participant's groups at all-time intervals. Nevertheless, for the primary outcome measurement, the assessor was the participant, so the study cannot properly be called assessor-blind either. Also, participants were asked not to discuss their treatment details with their peers or the outcome-measuring therapist. In addition, authors did not have access to information that could identify individual participants during or after data collection.

## Statistical methods

The normality of study participants' demographic characteristics was analyzed through the Kolmogorov-Smirnov test. The outcome data were presented in the form of a mean and standard deviation with a 95% confidence interval. The study followed the intention to treat the principle method by including the participants' missing follow-up data in the data analysis who were randomized. The effects of treatment at different time intervals were analyzed using a $3 \times 4$ linear mixed model analysis (LMM), with the patient as a random factor and treatment groups; cervical spine manipulation, thoracic spine manipulation, and conventional physical therapy and time intervals; baseline, four weeks, eight weeks, and at six months) as fixed factors. Following 3x4 LMM model analysis, post hoc Bonferroni analysis was done to compare

the three study groups pair-wise. For all the statistical tests a statistical significance level of α = 0.05 was set and the software GraphPad-Prism (version 9.1), Boston, MA, USA was used for analysis.

## Results

### Participants

Out of the 130 participants screened, eight had a VAS score greater than 8, ten participants had some sort of orthopaedic injuries, four participants had undergone joint surgeries, and twelve refused to be involved in the research and were excluded. N = 96 participants were chosen based on the eligibility criteria and allocated to one of the three groups. Two participants in the CSM and CPT groups, and three in the TSM group, did not complete the 4-week treatment program with a 6-month follow-up (Fig 1). Compliance with follow-up data collection at 6 months was 93%, adherence to study protocols (e.g., number of visits) was 100%, and none of the participants in the three groups received any additional care that was not included in the three study interventions. In all three groups, females (53–56%) are affected more than males. The clinical presentation of headache is more unilateral (78% - 84%) than bilateral, and the majority of CGH cases have associated neck pain (84% - 88%). (Table 1).

### Primary outcome

The mean and standard deviation (SD) of the CgH frequency score between the three groups at four-time period are shown in Tables 2 and 3. Over 4 weeks of different interventions, there is a significant change in CgH pain frequency level between the CSM (7.9; CI 95% 7.41 to 8.38), and TSM (4.7 CI 95% 4.21 to 5.18), groups (p = 0.001). A similar improvement can be seen in 8-weeks and at 6-months' measurement. The post-hoc Bonferroni analysis and the standard mean difference showed more percentage of improvement in CgH pain frequency in

**Table 1. Demographic details of CSM, TSM and CPT groups.**

| Variable | | CSM | TSM | CPT |
|---|---|---|---|---|
| Age (year) | - | 35.6 ± 3.8 | 34.8 ± 3.2 | 36.2 ± 3.7 |
| Gender | | | | |
| | Male | 14 (44%) | 15 (47%) | 14 (44%) |
| | Female | 18 (56%) | 17 (53%) | 18 (56%) |
| Height (cm) | - | 164.6 ± 3.8 | 163.5 ± 4.1 | 165.3 ± 4.4 |
| Weight (kg) | - | 72.92 ± 4.2 | 71.21 ± 4.6 | 73.83 ± 4.5 |
| BMI (kg/m$^2$) | - | 24.3 ± 2.23 | 24.5 ± 1.92 | 23.8 ± 2.01 |
| CgH duration (year) | - | 6.6 ± 2.9 | 6.8 ± 3.1 | 5.9 ± 3.2 |
| CgH frequency (per day) | - | 0.74 ± 0.15 | 0.72 ± 0.11 | 0.69 ± 0.12 |
| CgH intensity(0–10) | - | 6.9 ± 1.4 | 7.2 ± 1.3 | 7.3 ± 1.3 |
| Headache | | | | |
| | Unilateral | 26 (81%) | 25 (78%) | 27 (84%) |
| | Bilateral | 6 (19%) | 7 (12%) | 5 (16%) |
| Neck pain | | | | |
| | Yes | 28 (88%) | 27 (84%) | 28 (88%) |
| | No | 4 (12%) | 5 (16%) | 4(12%) |

CSM–Cervical spine manipulation, TSM–Thoracic spine manipulation, CPT- Conventional physiotherapy, BMI–Body mass index, CgH–Cervicogenic Headache.

**Table 2. Mean ± SD outcome measures of CSM, TSM and CPT groups.**

| Variable | Time | CSM | TSM | CPT |
|---|---|---|---|---|
| CgH Frequency (no of days per 4 weeks) | Baseline | 16.8 ± 1.8 | 17.2 ± 1.9 | 17.4 ± 1.7 |
| | 4 weeks | 11.2 ± 1.4 | 14.1 ± 1.6 | 15.8 ± 1.4 |
| | 8 weeks | 6.2 ± 0.9 | 9.5 ± 0.9 | 13.4 ± 1.1 |
| | 6 months | 2.9 ± 0.5 | 6.1 ± 0.7 | 10.8 ± 1.1 |
| CgH Pain intensity (0–10) | Baseline | 7.2 ± 0.8 | 6.8 ± 0.7 | 7.1 ± 0.7 |
| | 4 weeks | 4.1 ± 0.5 | 5.2 ± 0.5 | 6.2 ± 0.5 |
| | 8 weeks | 2.8 ± 0.4 | 4.1 ± 0.4 | 5.1 ± 0.4 |
| | 6 months | 0.8 ± 0.2 | 1.7 ± 0.3 | 3.7 ± 0.4 |
| CgH Disability | Baseline | 57.88 ± 6.5 | 57.21 ± 6.8 | 56.91 ± 5.9 |
| | 4 weeks | 45.73 ± 5.5 | 49.38 ± 5.6 | 52.38 ± 5.3 |
| | 8 weeks | 36.41 ± 4.3 | 41.73 ± 5.1 | 49.67 ± 4.5 |
| | 6 months | 31.19 ± 3.8 | 39.54 ± 4.5 | 48.37 ± 4.1 |
| Neck pain frequency (no of days per 4 weeks) | Baseline | 23.9 ± 3.2 | 23.6 ± 3.4 | 22.9 ± 3.6 |
| | 4 weeks | 15.8 ± 2.2 | 18.5 ± 2.4 | 20.4 ± 2.3 |
| | 8 weeks | 9.3 ± 1.6 | 12.3 ± 1.8 | 18.2 ± 1.5 |
| | 6 months | 3.4 ± 0.4 | 8.2 ± 0.9 | 16.1 ± 1.5 |
| Neck pain intensity (0–10) | Baseline | 7.2 ± 0.6 | 7.1 ± 0.6 | 6.9 ± 0.5 |
| | 4 weeks | 4.8 ± 0.5 | 6.2 ± 0.6 | 6.1 ± 0.5 |
| | 8 weeks | 2.6 ± 0.4 | 4.2 ± 0.5 | 5.1 ± 0.4 |
| | 6 months | 0.6 ± 0.2 | 1.9 ± 0.4 | 3.7 ± 0.4 |
| Neck pain threshold | Baseline | 262.1 ± 21.6 | 261.3 ± 22.3 | 262.7 ± 22.5 |
| | 4 weeks | 268.5 ± 20.2 | 266.3 ± 21.8 | 263.5 ± 21.6 |
| | 8 weeks | 274.5 ± 18.3 | 272.4 ± 19.4 | 267.8 ± 19.6 |
| | 6 months | 288.2 ± 17.3 | 278.3 ± 18.2 | 269.5 ± 17.3 |
| Flexon rotation test (Right side) | Baseline | 25.18 ± 7.4 | 25.12 ± 7.3 | 26.01 ± 7.2 |
| | 4 weeks | 31.76 ± 6.3 | 30.19 ± 6.3 | 27.12 ± 6.5 |
| | 8 weeks | 39.73 ± 5.6 | 35.16 ± 5.4 | 31.42 ± 6.1 |
| | 6 months | 45.21 ± 5.3 | 38.22 ± 5.1 | 32.31 ± 5.8 |
| Flexon rotation test (Left side) | Baseline | 24.93 ± 6.2 | 24.72 ± 6.1 | 24.88 ± 6.3 |
| | 4 weeks | 33.12 ± 5.9 | 27.93 ± 5.9 | 26.63 ± 5.8 |
| | 8 weeks | 37.81 ± 5.2 | 32.53 ± 5.5 | 29.92 ± 5.1 |
| | 6 months | 44.23 ± 5.1 | 36.23 ± 5.1 | 31.92 ± 4.9 |
| Neck Disability Index (0–100 with 100 worst) | Baseline | 51.01 ± 11.1 | 51.25 ± 10.8 | 50.97 ± 11.2 |
| | 4 weeks | 37.36 ± 9.2 | 43.34 ± 9.8 | 45.32 ± 10.2 |
| | 8 weeks | 24.83 ± 6.1 | 31.32 ± 6.2 | 38.53 ± 6.9 |
| | 6 months | 11.28 ± 3.4 | 23.56 ± 5.4 | 34.92 ± 6.3 |
| Quality of life (EQ-5D) (0–100 with 100 best) | Baseline | 24.9 ± 4.5 | 25.1 ± 3.9 | 24.8 ± 4.2 |
| | 4 weeks | 48.3 ± 4.8 | 36.3 ± 4.8 | 34.9 ± 4.3 |
| | 8 weeks | 64.5 ± 5.5 | 55.3 ± 5.1 | 40.1 ± 4.6 |
| | 6 months | 79.2 ± 6.2 | 62.4 ± 5.9 | 46.7 ± 4.9 |

CSM–Cervical spine manipulation, TSM–Thoracic spine manipulation, CPT- Conventional physiotherapy, CgH–Cervicogenic Headache.

the CSM group than TSM and CPT groups (Fig 5A). The complete interpretation shows a slight leaning towards the CSM group with (MCID = 7.9) than the TSM and CPT group in CgH frequency at 6 months' follow-up. Also, the effect size Cohen's (d = 9.8) shows a greater effect in the CSM group than TSM and CPT groups.

**Table 3. Between-group comparisons presented as mean differences (1st: CSM × TSM; 2nd: CSM × CPT; 3rd: TSM × CPT).**

| Variable / Time | | 4 weeks | 8 weeks | 6 months |
|---|---|---|---|---|
| | | **Mean difference 95% (upper limit–lower limit)** | | |
| **CgH Frequency** | **CSM × TSM** | 2.9 (2.02 to 3.77) | 3.3 (2.72 to 3.87) | 3.2 (2.71 to 3.68) |
| | **P–value** | 0.001, d = 1.93 | 0.001, d = 0.36 | 0.001, d = 5.33 |
| | **CSM × CPT** | 4.6 (3.72 to 5.47) | 7.2 (6.62 to 7.77) | 7.9 (7.41 to 8.38) |
| | **P–value** | 0.001, d = 3.28 | 0.001, d = 7.20 | 0.001, d = 9.80 |
| | **TSM × CPT** | 1.7 (0.82 to 2.57) | 3.9 (3.32 to 4.47) | 4.7 (4.21 to 5.18) |
| | **P–value** | 0.001, d = 1.13 | 0.001, d = 3.90 | 0.004, d = 5.22 |
| **CgH Pain Intensity** | **CSM × TSM** | 1.1 (0.80 to 1.39) | 1.3 (1.06 to 1.53) | 0.9 (0.71 to 1.08) |
| | **P–value** | 0.001, d = 2.2 | 0.001, d = 0.32 | 0.001, d = 3.6 |
| | **CSM × CPT** | 2.1 (1.80 to 2.39) | 2.3 (2.06 to 2.53) | 2.9 (2.71 to 3.08) |
| | **P–value** | 0.001, d = 4.2 | 0.003, d = 5.75 | 0.874 d = 9.66 |
| | **TSM × CPT** | 1.0 (0.70 to 1.29) | 1.0 (0.76 to 1.23) | 2.0 (1.81 to 2.18) |
| | **P–value** | 0.001, d = 2.0 | 0.001, d = 2.5 | 0.108, d = 5.71 |
| **CgH Disability** | **CSM × TSM** | 3.6 (0.39 to 6.90) | 5.3 (2.55 to 8.08) | 8.3 (5.88 to 10.81) |
| | **P–value** | 0.024, d = 0.65 | 0.001, d = 1.12 | 0.001, d = 2.01 |
| | **CSM × CPT** | 6.6 (3.39 to 9.90) | 13.2 (10.49 to 16.02) | 17.1 (14.7 to 19.6) |
| | **P–value** | 0.001, d = 1.23 | 0.001, d = 3.01 | 0.001, d = 4.34 |
| | **TSM × CPT** | 3.00 (-0.25 to 6.25) | 7.9 (5.17 to 10.70) | 8.8 (6.3 to 11.2) |
| | **P–value** | 0.774, d = 0.55 | 0.001, d = 1.65 | 0.001, d = 2.05 |
| **Neck pain frequency** | **CSM × TSM** | 2.7 (1.32 to 4.07) | 3.0 (2.02 to 3.97) | 4.8 (4.18 to 5.41) |
| | **P–value** | 0.001, d = 1.17 | 0.001, d = 1.76 | 0.001, d = 7.38 |
| | **CSM × CPT** | 4.6 (3.22 to 5.97) | 8.9 (7.92 to 8.97) | 12.7 (12.08 to 13.31) |
| | **P–value** | 0.001, d = 2.04 | 0.002, d = 2.04 | 0.874, d = 13.36 |
| | **TSM × CPT** | 1.9 (0.52 to 3.27) | 5.9 (4.92 to 6.87) | 7.9 (7.28 to 8.51) |
| | **P–value** | 0.003, d = 0.80 | 0.001, d = 3.57 | 0.746, d = 6.58 |
| **Neck pain intensity** | **CSM × TSM** | 1.4 (1.08 to 1.71) | 1.6 (1.34 to 1.85) | 1.3 (1.09 to 1.50) |
| | **P–value** | 0.001, d = 2.54 | 0.001, d = 3.55 | 0.001, d = 4.33 |
| | **CSM × CPT** | 1.3 (0.98 to 1.61) | 2.5 (2.24 to 2.75) | 3.1 (2.89 to 3.30) |
| | **P–value** | 0.001, d = 2.60 | 0.001, d = 6.25 | 0.873, d = 10.3 |
| | **TSM × CPT** | -0.1 (-0.41 to 0.21) | 0.9 (0.64 to 1.15) | 1.8 (1.59 to 2.00) |
| | **P–value** | 0.736, d = 0.18 | 0.001, d = 2.0 | 0.001, d = 4.5 |
| **Neck pain threshold** | **CSM × TSM** | -2.2 (-14.83 to 10.43) | -2.1 (-13.47 to 9.27) | -9.9 (-20.38 to 0.58) |
| | **P–value** | 0.909, d = 0.10 | 0.899, d = 0.11 | 0.068, d = 0.55 |
| | **CSM × CPT** | -5 (-17.63 to 7.63) | -6.7 (-18.07 to 4.67) | -18.7 (-29.18 to -8.21) |
| | **P–value** | 0.614, d = 0.23 | 0.343, d = 0.35 | 0.001, d = 1.08 |
| | **TSM × CPT** | -2.8 (15.43 to 9.83) | -4.6 (-15.97 to 6.77) | -8.8 (-19.28 to 1.68) |
| | **P–value** | 0.857, d = 0.12 | 0.602, d = 0.23 | 0.118, d = 0.49 |
| **Flexion rotation test (Right side)** | **CSM × TSM** | -1.5 (-5.36 to 2.22) | -4.5 (-7.9 to -1.1) | -6.9 (-10.21 to -3.76) |
| | **P–value** | 0.587, d = 0.24 | 0.005, d = 0.83 | 0.001, d = 1.34 |
| | **CSM × CPT** | -4.64 (-8.43 to -0.84) | -8.3 (-11.7 to -4.91) | -12.9 (-16.12 to -9.67) |
| | **P–value** | 0.012, d = 0.72 | 0.001, d = 1.42 | 0.001, d = 2.32 |
| | **TSM × CPT** | -3.07 (-6.86 to 0.72) | -3.74 (-7.13 to -0.34) | -5.91 (-9.13 to -2.68) |
| | **P–value** | 0.136, d = 0.47 | 0.027, d = 0.65 | 0.001, d = 1.08 |
| **Flexion rotation test (Left side)** | **CSM × TSM** | -5.19 (-8.68 to -1.69) | -5.2 (-8.41 to -2.14), | -8.0 (-10.99 to -5.00) |
| | **P–value** | 0.001, d = 0.87 | 0.001, d = 1.00 | 0.001, d = 1.56 |
| | **CSM × CPT** | -6.4 (-9.98 to -2.99) | -7.8 (-11.02 to -4.75) | -12.31 (-15.30 to -9.31) |
| | **P–value** | 0.001, d = 1.10 | 0.001, d = 1.53 | 0.001, d = 2.46 |
| | **TSM × CPT** | -1.3 (-4.79 to 2.19) | -2.61 (-5.74 to 0.52) | -4.31 (-7.30 to -1.31) |
| | **P–value** | 0.650, d = 0.22 | 0.122, d = 0.49 | 0.002, d = 0.86 |

*(Continued)*

**Table 3.** (Continued)

| Variable / Time | | 4 weeks | 8 weeks | 6 months |
|---|---|---|---|---|
| | | **Mean difference 95% (upper limit–lower limit)** | | |
| **Neck Disability Index** | **CSM × TSM** | 5.98 (0.17 to 11.78) | 6.49 (2.67 to 10.30) | 12.28 (9.19 to 15.36) |
| | **P–value** | 0.041, d = 0.62 | 0.001, d = 1.05 | 0.001, d = 2.79 |
| | **CSM × CPT** | 7.96 (2.15 to 13.76) | 13.7 (9.88 to 17.51) | 23.64 (20.55 to 26.72) |
| | **P–value** | 0.004, d = 0.82 | 0.001, d = 1.43 | 0.001, d = 4.87 |
| | **TSM × CPT** | 1.98 (-3.82 to 7.78) | 7.21 (3.39 to 11.02) | 11.36 (8.27 to 14.44) |
| | **P–value** | 0.696, d = 0.19 | 0.001, d = 1.10 | 0.001, d = 1.95 |
| **Quality of life** | **CSM × TSM** | -12.0 (-14.76 to -9.23) | -9.2 (-12.2 to -6.1) | -16.8 (-20.19 to -13.40) |
| | **P–value** | 0.001, d = 2.5 | 0.001, d = 1.73 | 0.001, d = 2.77 |
| | **CSM × CPT** | -13.4 (-16.16 to -10.63) | -24.4 (-27.4 to -21.37) | -32.5 (-35.89 to -29.10) |
| | **P–value** | 0.001, d = 2.94 | 0.001, d = 4.83 | 0.002, d = 5.85 |
| | **TSM × CPT** | -1.4 (-4.16 to 1.36) | -15.20 (-18.22 to -12.17) | -15.7 (-19.09 to -12.30) |
| | **P–value** | 0.452, d = 0.30 | 0.001, d = 5.85 | 0.001, d = 2.90 |

CSM–Cervical spine manipulation, TSM–Thoracic spine manipulation, CPT- Conventional physiotherapy, CgH–Cervicogenic Headache, d–Cohen's d (effect size).

## Secondary outcomes

The mean and standard deviation (SD) of the secondary outcomes between the three groups at four-time period are shown in Tables 2 and 3. After 4 weeks of intervention, there are statistically significant variations in CgH pain intensity, CgH disability, NP (frequency, intensity, and threshold), CFRT (Right and left), NDI, and QoL score between the CSM (2.9; CI 95% 2.71 to 3.08), (17.1; CI 95% 14.7 to 19.6) (12.7; CI 95% 12.08 to 13.31), (3.1; CI 95% 2.89 to 3.30), (-18.7; CI 95% -29.18 to -8.21), (-12.9, -12.31; CI 95% -16.12 to -9.67, -15.30 to -9.31), (23.64; CI 95% 20.55 to 26.72), (-32.5; CI 95% -35.89 to -29.10) and TSM (2.0; CI 95% 1.81 to 2.18), (8.8; CI 95% 6.3 to 11.2), (7.9; CI 95% 7.28 to 8.51), (1.8; CI 95% 1.59 to 2.00), (-8.8; CI 95% -19.28 to 1.68), (-5.91, -4.31; CI 95% -9.13 to -2.68, -7.30 to -1.31), (11.36; CI 95% 8.27 to 14.44), (-15.7; CI 95% -19.09 to -12.30) groups (p<0.001) respectively. The post hoc Bonferroni analysis and the standard mean difference showed more percentage of improvement in the secondary outcomes in the CSM group than TSM and CPT groups (Fig 5A and 5B). The complete interpretation shows a slight leaning towards the CSM group in MCID scores of CgH pain intensity = 2.9, CgH disability = 17.18, NP (frequency = 12.7, intensity = 3.1, and threshold = 18.7), CFRT (Right = 12.9 and left = 12.31), NDI = 23.64, and QoL = 32.5 than TSM and CPT group in all the secondary variables at 6 months' follow-up. Similarly, the effect size Cohen's d for CgH pain intensity = 11.6, CgH disability = 4.34, NP (frequency = 13.36, intensity = 10.3, and threshold = 1.08), CFRT (Right = 2.32 and left = 2.46), NDI = 4.87, and QoL = 5.85 shows greater effect than TSM and CPT groups.

## Discussion

This was the first powered randomized trial to compare the effects of cervical and thoracic manipulation for patients with CgH. Cervical manipulation was found to be superior to thoracic manipulation and conventional PT (massage) for improving days with CgH, as well as headache and neck pain and disability, to 6 months. According to this study, after four weeks of intervention the cervical spine manipulation (CSM) group showed statistically significant changes in all the outcome measures in CgH patients. When compared to TSM and conventional physical therapy, CSM is more effective in reducing the primary outcome CgH

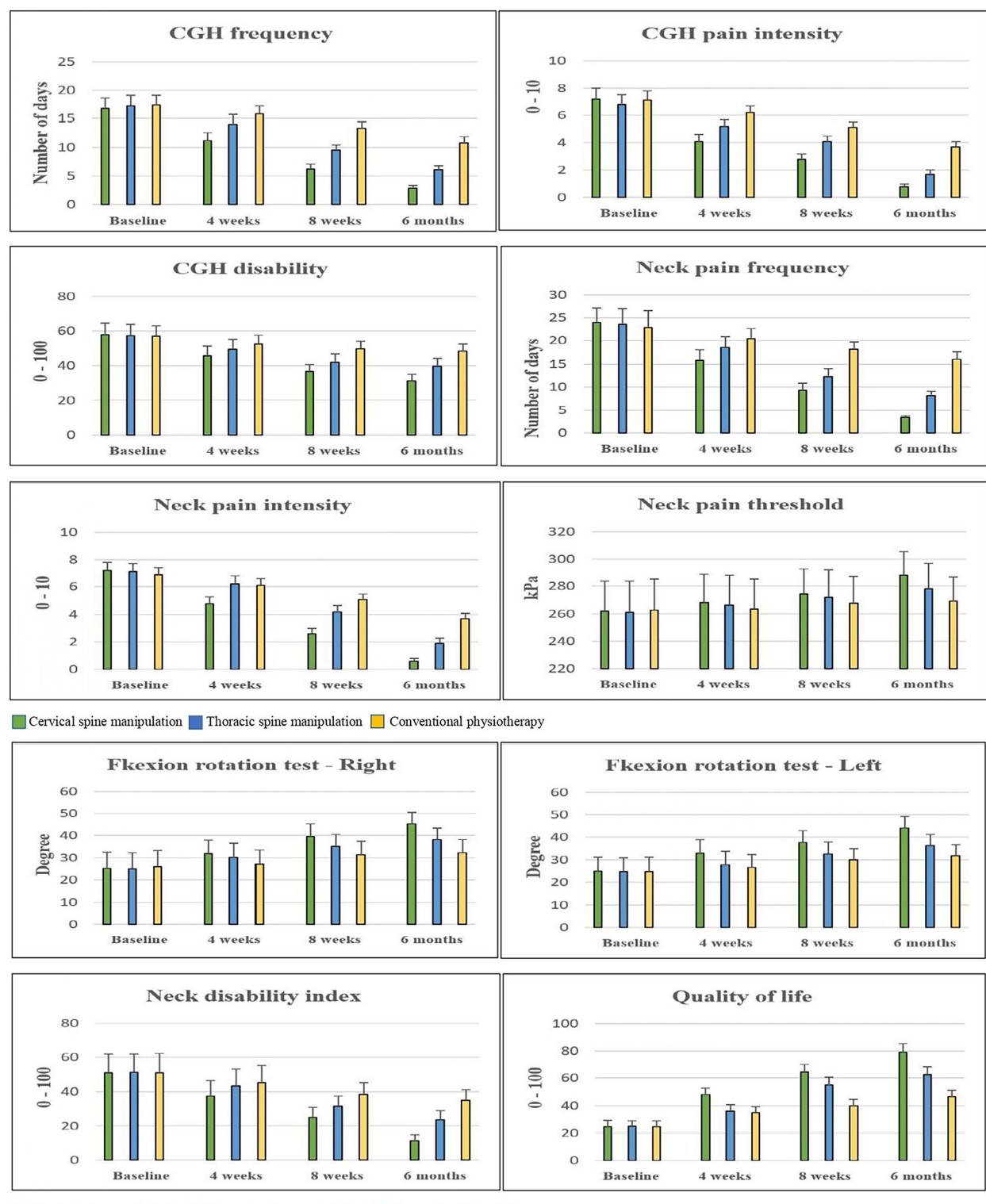

**Fig 5.** a. Pre and post-outcome measures of CSM, TSM and CPT groups. b. Pre and post-outcome measures of CSM, TSM and CPT groups.

frequency, which was clinically important. The same improvements were noted in all the other outcome measures such as CgH pain, neck pain frequency, neck pain intensity, neck pain threshold, range of motion, and quality of life and these reports were supported by Dunning JR et al study [16]. So far, no studies have looked at the changes in CgH disability and neck disability, but this study looked at the benefits of cervical spine manipulation over thoracic spine manipulation in CgH patients. However, these findings were contradictory to the observation of Borusiak et al and found that there is no significant difference comparing the manipulation groups with placebo groups concerning the defined main outcome measures [30].

The mechanism of action has yet to be determined. Manipulation of the cervical spine may promote afferent nerve fiber activity through stimulation of the cervical joint receptors. It may improve the overall action and properties of the neck muscles by activating the alpha motor neuron [31]. It alters the sensory fiber activity by activating the joint receptors, thereby changing the α-motor neuron activity levels and subsequent muscle reaction. Because of the high mobility of the cervical spine, CSM can stimulate the receptors of deep neck muscles and sub-occipital muscles, which TSM is not able to do [32]. Other theories for the pain-modulating effects of cervical manipulation included biomechanical, vertebral (temporal summation), and neural (central descending pain inhibitory pathway) mechanisms which were noted by Bialosky JE et al and Haavik-Taylor H et al [6,33,34].

Thoracic manipulation was also found to be more effective than conventional PT in improving both the primary and secondary outcomes. TSM was also helpful in improving the pain parameters, functional disability and quality of life significantly. It was supported by McDevitt AW et al [18] and suggests that TSM had no effect on headache-related disability but resulted in significant improvements in neck-related disability and participants reported perceived improvement. Although several types of research have been conducted to find the effects of different types of spinal manipulation, the real mechanical and neuro-physiological alterations behind these changes have not yet been found and are unclear, but Bialosky et al. suggest that the effects may be due to potential neurophysiological and biomechanical effects, as well as possibly placebo effect [33]. According to Suvarnnato T et al, the neurophysiologic response of pain reduction in patients with chronic neck pain through TSM is that it induces a reflex inhibition of pain or muscle relaxation reflex by modifying the discharge of proprioceptive Group I and II afferents. It also activates descending inhibitory mechanisms resulting in pain reduction in distant areas from the manipulation. Through these mechanisms, the thoracic manipulation may induce ventral periaqueductal gray (vPAG) in the brain, which activates endogenous opioid peptides resulting in pain reduction in different areas [35]. The little changes in the conventional physical therapy group on pain intensity and other outcome variables explained the analgesic effect of CPT on cervicogenic headache. Application of massage on the trigeminal-cervical area reduces inflammatory responses, reduces neural sensitivity and plays a significant function in decreasing the tension of the sub-occipital and para-vertebral cervical spine muscles, which is another important mechanism of physical therapy on CgH patients [36,37]. The findings of this trial should assist physiotherapists in making decisions to select the best manual therapy approach for CgH patients.

## Limitations

The study had some limitations during its execution, which should be considered for future studies. First, the study included both genders, but the reports collected were not calculated independently during the statistical analysis, any possible gender differences may influence the research reports. Second, it is impossible to ensure that the subjects completed the questionnaires daily rather than after a week or four weeks. The standard for headache outcomes is a

headache diary that uses technology to ensure that frequency and pain intensity are reported immediately on the day stipulated, not added in later. Third, this study lacks a placebo group to determine the true effects of treatment groups. The beneficial effects of various manipulation techniques on pain and other symptoms in Cervicogenic headache were investigated. Finally, the treatment preference of physiotherapists and patients was not asked which could have affected the results due to clinical equipoise.

## Conclusion

The current randomized controlled trial found that cervical spine manipulation was more effective in improving pain parameters (intensity, frequency and threshold), functional disability and quality of life in patients with cervicogenic headache than thoracic spine manipulation and conventional physiotherapy. This study also adds to the evidence in the field of manual therapy for patients with CgH. Future studies are recommended to identify the biomechanical and biochemical mechanisms underlying the clinical and functional changes engendered by manipulation in the treatment of cervicogenic headache patients.

## Supporting information

**S1 Checklist. CONSORT 2010 checklist of information to include when reporting a randomised trial\*.**
(DOCX)

**S1 Data.**
(XLSX)

**S1 File.**
(DOCX)

## Author Contributions

**Conceptualization:** Gopal Nambi, Mshari Alghadier, Mudathir Mohamedahmed Eltayeb, Osama R. Aldhafian, Ayman K. Saleh, Nesreen Alsanousi, Alaa Jameel A. Albarakati, Mohamed A. Omar, Mohamed Nagah Ahmed Ibrahim, Abdehamid A. Attallah, Mohammed Abdelgwad Ismail, Mohamed Elfeshawy.

**Data curation:** Gopal Nambi, Mshari Alghadier, Mudathir Mohamedahmed Eltayeb, Osama R. Aldhafian, Ayman K. Saleh, Nesreen Alsanousi, Alaa Jameel A. Albarakati, Mohamed A. Omar, Mohamed Nagah Ahmed Ibrahim, Abdehamid A. Attallah, Mohammed Abdelgwad Ismail, Mohamed Elfeshawy.

**Formal analysis:** Gopal Nambi, Mshari Alghadier, Alaa Jameel A. Albarakati, Abdehamid A. Attallah.

**Funding acquisition:** Gopal Nambi, Mudathir Mohamedahmed Eltayeb, Osama R. Aldhafian, Ayman K. Saleh, Alaa Jameel A. Albarakati, Mohamed A. Omar, Mohamed Nagah Ahmed Ibrahim, Abdehamid A. Attallah, Mohammed Abdelgwad Ismail, Mohamed Elfeshawy.

**Investigation:** Mshari Alghadier, Osama R. Aldhafian, Ayman K. Saleh, Nesreen Alsanousi, Alaa Jameel A. Albarakati, Mohamed A. Omar, Mohamed Nagah Ahmed Ibrahim, Abdehamid A. Attallah, Mohammed Abdelgwad Ismail, Mohamed Elfeshawy.

**Methodology:** Gopal Nambi, Mshari Alghadier, Mudathir Mohamedahmed Eltayeb, Osama R. Aldhafian, Ayman K. Saleh, Nesreen Alsanousi, Alaa Jameel A. Albarakati, Mohamed Nagah Ahmed Ibrahim, Abdehamid A. Attallah.

**Project administration:** Gopal Nambi, Mudathir Mohamedahmed Eltayeb, Ayman K. Saleh, Nesreen Alsanousi, Mohamed A. Omar, Mohammed Abdelgwad Ismail, Mohamed Elfeshawy.

**Resources:** Mshari Alghadier, Osama R. Aldhafian, Nesreen Alsanousi, Mohamed A. Omar, Mohamed Nagah Ahmed Ibrahim, Mohammed Abdelgwad Ismail, Mohamed Elfeshawy.

**Software:** Mshari Alghadier, Mudathir Mohamedahmed Eltayeb, Ayman K. Saleh, Alaa Jameel A. Albarakati, Mohamed A. Omar, Mohamed Elfeshawy.

**Supervision:** Gopal Nambi, Mudathir Mohamedahmed Eltayeb, Osama R. Aldhafian, Mohamed A. Omar, Abdehamid A. Attallah, Mohammed Abdelgwad Ismail.

**Validation:** Mshari Alghadier, Mudathir Mohamedahmed Eltayeb, Nesreen Alsanousi, Alaa Jameel A. Albarakati, Mohamed A. Omar.

**Visualization:** Gopal Nambi, Ayman K. Saleh, Nesreen Alsanousi, Mohamed Nagah Ahmed Ibrahim, Abdehamid A. Attallah, Mohamed Elfeshawy.

**Writing – original draft:** Mshari Alghadier, Osama R. Aldhafian, Ayman K. Saleh, Nesreen Alsanousi, Alaa Jameel A. Albarakati, Mohamed Nagah Ahmed Ibrahim, Mohammed Abdelgwad Ismail, Mohamed Elfeshawy.

**Writing – review & editing:** Gopal Nambi, Mshari Alghadier, Mudathir Mohamedahmed Eltayeb, Osama R. Aldhafian, Nesreen Alsanousi, Alaa Jameel A. Albarakati, Mohamed A. Omar, Mohamed Nagah Ahmed Ibrahim, Abdehamid A. Attallah, Mohammed Abdelgwad Ismail.

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
