## [Decision Letter · Decision Letter 0]

19 Sep 2023

PONE-D-23-12627Pragmatic effects of spinal thrust manipulations on pain parameters: cervical spine versus thoracic spine in Cervicogenic headache – A prospective, single-blinded, randomized controlled study.PLOS ONE

Dear Dr. Nambi,

Thank you for submitting your manuscript to PLOS ONE. After careful consideration, we feel that it has merit but does not fully meet PLOS ONE’s publication criteria as it currently stands. Therefore, we invite you to submit a revised version of the manuscript that addresses the points raised during the review process.

We look forward to receiving your revised manuscript.

Kind regards,

Jianhong Zhou

Staff Editor

PLOS ONE

Journal Requirements:

"This study is supported via funding from Prince Sattam bin Abdulaziz University project number (PSAU/2023/R/1444)."

"This study is supported via funding from Prince Sattam bin Abdulaziz University project number (PSAU/2023/R/1444)."

"This study is supported via funding from Prince Sattam bin Abdulaziz University project number (PSAU/2023/R/1444)."

Reviewers' comments:

Reviewer's Responses to Questions

**Comments to the Author**

1. Is the manuscript technically sound, and do the data support the conclusions?

Reviewer #1: Partly

Reviewer #2: Yes

2. Has the statistical analysis been performed appropriately and rigorously? 

Reviewer #1: Yes

Reviewer #2: Yes

3. Have the authors made all data underlying the findings in their manuscript fully available?

Reviewer #1: No

Reviewer #2: Yes

4. Is the manuscript presented in an intelligible fashion and written in standard English?

Reviewer #1: Yes

Reviewer #2: No

5. Review Comments to the Author

Reviewer #1: This is an interesting study and the protocol looks sound, however, the manipulation techniques, as described, do not look sound at all. Not contemporary manual therapy practice.

See my comments in the pdf file.

With some major revisions, this could be a great paper for publication.

Reviewer #2: This was a novel study that gives first evidence for a difference between cervical and thoracic manipulation for cervicogenic headache, as well as an advantage for manipulation over massage used in conventional physical therapy. On the whole, the study is well designed with appropriate. primary and secondary outcomes for CGH. There was excellent adherence to care and compliance with data collection.

There are some limitations that need to be considered. The sample size was small albeit the study was powered. The standard for headache outcomes is a headache diary that uses technology to ensure that frequency and pain intensity are reported immediately on the day stipulated, not added in later. Also, the conventional physical therapy intervention appears to have visits of longer duration than the manipulation visits which could have introduced attention bias and reduced treatment effect estimates. The manuscript needs work. More details and language editing are required.

Title:

1. Try “Comparative effectiveness of cervical vs thoracic spinal-thrust manipulation for care of cervicogenic headache: a randomized controlled trial”

Introduction

2. 1st paragraph: Give a citation for CGH having socioeconomic impact and being a significant public health burden.

3. Reference 13: This was a pilot study. A full-scale RCT on dose-response was published in 2018 in The Spine Journal. This would be a better reference.

Material and Methods

4. Design: Blinding: This study was not “single blind.” This designation would require that the participant or provider was blinded. The assessor was blinded for data collection, but for the primary outcome the assessor was the participant, so the study cannot properly be called assessor-blind either.

5. Participants: Include a reference for CONSORT.

6. Interventions: State that all three groups received ten minutes of hydrocollator heat prior to intervention (manipulation or massage).

7. Spinal Manipulation Therapy:

Include the reference number for “Peterson and Bergman.”

Was a “cracking” sound required for the manipulation to be considered successful?

8. Cervical Manipulation: Were multiple segment-specific manipulations permitted on each side based on segmental dysfunction or was a single general, nonspecific manipulation performed on each side?

9. CGH Disability: Briefly explain how the HIT questionnaire is scored: scale for each of the six items and direction of scoring, totaling the score, and total scale range.

10. Sample size: Why was a four day’s difference between groups in CGH frequency chosen for the sample size calculation? Is this a clinically important difference between groups?

11. Analysis: Baseline comparisons:

Baseline statistical comparisons should not be performed. This is because in a randomized trial, baseline group differences are attributable to chance by definition, due to random allocation of participants. Remove baseline comparisons from text and tables.

12. Analysis: LMM: Further explain the model used. Was this a random-intercept model? If appropriate, identify which covariance structure was used.

13. Analysis: Post hoc analysis: Include here that post hoc Bonferroni analysis was use following the omnibus 3x4 LMM model to compare the three study groups pair-wise.

14. Analysis: Intention-to-Treat:

Include this under Analysis, not Results Participants.

Explain how you define intention-to treat because there are different definitions. Were participants with missing follow-up data included in the analysis?

Results

15. Participants:

Baseline characteristics: Make a general statement that the three groups were uniform in baseline characteristics and baseline outcome variables.

CGH was defined as unilateral, yet there were participants with bilateral headache. Explain why these were included?

16. Partial eta-squared: Remove this statistic from the text and tables. This effect size statistic is uncommon in biomedical research (usually found in the social sciences literature). It is also not defined or interpreted in this manuscript, so has no use.

17. Table 2: Show score range for NDI (0-100 with 100 worst) and EQ-5D (0-100 with 100 best) in the second column.

18. Table 3: CGH frequency: There are errors in the p-values for CSM x CPT at 8 weeks and 6 months.

Discussion

19. Organization: I suggest you start by stating this is the first powered study to compare cervical and thoracic manipulation for CGH and that the study shows that cervical manipulation was superior to thoracic manipulation and conventional PT (massage). Then compare studies and speculate about mechanisms.

20. 2nd Paragraph: This paragraph is confusing.

Statistically significant changes: This means a within-groups analysis of change over time. No within-groups analysis is identified under Methods or reported under Results. I think you are talking about between groups differences in improvement favoring CSM over TSM and CPT.

You report mean differences between groups from Table 3, not MCID. You can say that the differences between groups are clinically important, if you can give references that supports such claims. Also, start with the primary outcome, CGH frequency. Then discuss pain.

This study did not investigate thrust duration. You do not know if the thrust is less than 2ms in this study.

Perhaps change this paragraph into two separate paragraphs: one discussing clinical outcomes and the second discussing mechanism and theory.

21. Limitations: Differences in the length of study visits between groups adds attention bias and could reduce differences between manipulation groups and the massage group.

Conclusion

22. The conclusion in the abstract is more accurate and better written. You can use it again here and add a sentence or so to recommend a future research trajectory.

23. Strong evidence: This is a technical term used in systematic reviews. Strong evidence usually means multiple, high-quality trials with similar results.

6. PLOS authors have the option to publish the peer review history of their article (what does this mean?). If published, this will include your full peer review and any attached files.

Reviewer #1: **Yes: **Emilio "Louie" Puentedura

Reviewer #2: No

---

## [Author Response · Author response to Decision Letter 0]

4 Nov 2023

Dear Editor and reviewers, 

Thank you for your recent correspondence regarding our paper on ‘Comparative effectiveness of cervical vs thoracic spinal-thrust manipulation for care of cervicogenic headache: a randomized controlled trial.” We would like to thank you for the opportunity to submit a revised manuscript. We appreciate all the comments, suggestions, and positive criticism raised by the Editor and the reviewers which were of great value, and certainly helped us to improve the quality of our manuscript. We made all possible efforts to properly address and/or reply to the comments raised. Additionally, a complete review of the article was done for grammar and readability. A significant number of modifications were made to the article after several reviews. No large disagreements were made with the reviewer’s comments. The authors of this paper greatly thank each of the reviewers for spending time and provided such a constructive comments and fantastic input. The authors greatly appreciate the work of all the reviewers as well as the editors on this manuscript. Many points were a great addition to the quality of this article. Here is a corrected manuscript with a point-by-point response to comments from the reviewers. If any additional corrections can be corrected in the further rounds of revisions.

Regards

Dr. Gopal

---

## [Decision Letter · Decision Letter 1]

1 Dec 2023

PONE-D-23-12627R1Comparative effectiveness of cervical vs thoracic spinal-thrust manipulation for care of cervicogenic headache: a randomized controlled trial.PLOS ONE

Dear Dr. Nambi,

Thank you for submitting your manuscript to PLOS ONE. After careful consideration, we feel that it has merit but does not fully meet PLOS ONE’s publication criteria as it currently stands. Therefore, we invite you to submit a revised version of the manuscript that addresses the points raised during the review process.

We look forward to receiving your revised manuscript.

Kind regards,

André Pontes-Silva

Academic Editor

PLOS ONE

Additional Editor Comments:

Journal: PLOS ONE.

Title: Comparative effectiveness of cervical vs thoracic spinal-thrust manipulation for care of cervicogenic headache: a randomized controlled trial.

ID: PONE-D-23-12627R1.

Dear authors, thank you for your answers. You have compared the clinical effects of cervical over thoracic spine manipulation and conventional physiotherapy in patients with CgH. Please, when you review the article, in addition to highlighting the helped text, you must inform the page (and lines) of each of the adjustments. This will help you to get an opinion quickly.

—Regarding minimum clinically important difference (MCID), please report the MCID for each of your outcomes and report it in the results. Outcomes that do not have an MCID established in the literature should be highlighted in the study limitations.

—Please report the effect size value for all comparisons performed—the p-value does not indicate clinical significance (https://pubmed.ncbi.nlm.nih.gov/36325112/). You should calculate the effect size using Cohen's d-value for quantitative variables; and using Cohen's w-value for % (use this online calculator: https://www.psychometrica.de/effect_size.html). By the way, this article may help you classify d and w values of the effect size: https://pubmed.ncbi.nlm.nih.gov/37971135/.

—Table 1: The reported p-value refers to which comparison? This table needs to present 3 p-values; where are they? First p-value: CSM x TSM; Second p-value: CSM x CPT; third p-value: TSM x CPT. Remember to also report the effect size for each of these comparisons. Furthermore, correct the terms weight (the correct term is body mass [kg]) and height (the correct term is stature [cm or m]).

—Table 2: The p-value reported refers to which comparison (CSM x TSM, CSM x CPT, TSM x CPT)? This table also needs to present 3 p-values; where are they? In the legend you must detail the information in the table, inform the tests that generated the p-value, and the alpha established for statistical significance. Tables must be intuitive (remember to also inform the effect size for each of these comparisons).

—Table 3: In addition to the p-value, report the effect size for each of these comparisons.

Reviewers' comments:

Reviewer's Responses to Questions

**Comments to the Author**

1. If the authors have adequately addressed your comments raised in a previous round of review and you feel that this manuscript is now acceptable for publication, you may indicate that here to bypass the “Comments to the Author” section, enter your conflict of interest statement in the “Confidential to Editor” section, and submit your "Accept" recommendation.

Reviewer #1: All comments have been addressed

Reviewer #2: (No Response)

2. Is the manuscript technically sound, and do the data support the conclusions?

Reviewer #1: Partly

Reviewer #2: Yes

3. Has the statistical analysis been performed appropriately and rigorously? 

Reviewer #1: Yes

Reviewer #2: Yes

4. Have the authors made all data underlying the findings in their manuscript fully available?

Reviewer #1: Yes

Reviewer #2: Yes

5. Is the manuscript presented in an intelligible fashion and written in standard English?

Reviewer #1: No

Reviewer #2: No

6. Review Comments to the Author

Reviewer #1: While most of the comments have been adequately addressed. There are still some concerns. Please see attached pdf file. Also, the images of the CSM and TSM are not great. Cringe level = 8/10.

Reviewer #2: This manuscript has been greatly improved by the authors.

In the process, however, several changes that the authors said they made are missing from the revised manuscript.

I also make some suggestions for language improvements to the revisions in the Discussion and Conclusions.

Material and Methods

1. Sample size: State that 4 CgH days is considered the MCID between groups and include a citation.

2. Analysis: Baseline comparisons (Comment 11 from the first review):

The authors state that they removed the statistical comparisons of baseline variables between groups. This is not the case. Statements of no significance and/or p-values still appear in the following and these must be removed:

Abstract: Results: First sentence.

Results: Participants: Two sentences on demographics and clinical variables. (The new last sentence is supposed to replace the statistical comparisons with p-values).

Table 1: last column.

Table 3: Baseline column

3. Analysis: Intention-to-Treat:

This has parts:

You added that all patients were included in the analysis even if data were missing.

Also add that participants were analyzed in the group to which they were randomized.

Results

4. Partial eta-squared: (comment 16 from first review):

As per the response to reviewer comments, partial eta-squared has been removed from the text. However, it remains to be removed from Table 2.

5. Table 2: (Comment 17 from first review):

This has not been done as stated by the authors in the response to reviewer comments.

“Show score range for NDI (0-100 with 100 worst) and EQ-5D (0-100 with 100 best) in the second column.”

Discussion

6. 1st Paragraph:

Start the paragraph like this: “This was the first powered randomized trial to compare the effects of cervical and thoracic manipulation for patients with CgH. Cervical manipulation was found to be superior to thoracic manipulation and conventional PT (massage) for improving days with CgH, as well as headache and neck pain and disability, to 6 months.”

Make sure to check the grammar.

Briefly elaborate the contradiction with Borusiak et al. What did these authors find.

7. 2nd Paragraph:

Start the paragraph like this: “The mechanism of action has yet to be determined. Manipulation of the cervical spine may promote afferent nerve fiber activity through stimulation of the cervical joint receptors. It may improve…”

8. 3rd Paragraph:

Start the paragraph like this: “Thoracic manipulation was also found to be more effective than conventional PT in improving both the primary and secondary outcomes.”

Conclusion

9. Try this wording: “The current randomized controlled trial found that cervical spine manipulation was more effective in improving pain parameters (intensity, frequency and threshold), functional disability and quality of life in patients with cervicogenic headache than thoracic spine manipulation and conventional physiotherapy. Future studies are recommended to identify the biomechanical and biochemical mechanisms underlying the clinical and functional changes engendered by manipulation in the treatment of CgH.”

7. PLOS authors have the option to publish the peer review history of their article (what does this mean?). If published, this will include your full peer review and any attached files.

Reviewer #1: No

Reviewer #2: No

---

## [Author Response · Author response to Decision Letter 1]

17 Jan 2024

Editor comments:

Dear authors, thank you for your answers. You have compared the clinical effects of cervical over thoracic spine manipulation and conventional physiotherapy in patients with CgH. Please, when you review the article, in addition to highlighting the helped text, you must inform the page (and lines) of each of the adjustments. This will help you to get an opinion quickly.

Author response: Thank you for the comments and we have carefully considered all your comments and corrected the revised manuscript.

Editor comment 1: —Regarding minimum clinically important difference (MCID), please report the MCID for each of your outcomes and report it in the results. The study limitations should highlight outcomes that do not have an MCID established in the literature.

Author response: Thank you for the comments and as per your suggestion the MCID values of all the outcome measures were added. 

Editor comment 2: —Please report the effect size value for all comparisons performed—the p-value does not indicate clinical significance (https://pubmed.ncbi.nlm.nih.gov/36325112/). You should calculate the effect size using Cohen's d-value for quantitative variables; and using Cohen's w-value for % (use this online calculator: https://www.psychometrica.de/effect_size.html). By the way, this article may help you classify the d and w values of the effect size: https://pubmed.ncbi.nlm.nih.gov/37971135/.

Author response: Thank you for the comments and as per your suggestion the effect size Cohen’s d values of all the outcome measures were added. 

Editor comment 3: —Table 1: The reported p-value refers to which comparison? This table needs to present 3 p-values; where are they? First p-value: CSM x TSM; Second p-value: CSM x CPT; third p-value: TSM x CPT. Remember to also report the effect size for each of these comparisons. Furthermore, correct the terms weight (the correct term is body mass [kg]) and height (the correct term is stature [cm or m]).

Author response: Thank you for the comments and as per your suggestion this issue was discussed with our research statistician. He suggested that this is the demographic and clinical characteristics of the study participants’ analysis between the three groups. As per his suggestion, the effect size is not required for the above in this table. Also, the terms weight and height were modified in table 1.

Editor comment 4: —Table 2: The p-value reported refers to which comparison (CSM x TSM, CSM x CPT, TSM x CPT)? This table also needs to present 3 p-values; where are they? In the legend you must detail the information in the table, inform the tests that generated the p-value, and the alpha established for statistical significance. Tables must be intuitive (remember to also inform the effect size for each of these comparisons).

Author response: Thank you for the comments and as per your suggestion the p values for (CSM x TSM, CSM x CPT, TSM x CPT) are mentioned in Table 3.

Editor comment 5: —Table 3: In addition to the p-value, report the effect size for each of these comparisons.

Author response: Thank you for the comments and as per your suggestion the effect size for each of these comparisons was mentioned in Table 3.

Reviewer #1: 

Author response: Thank you for the comments and we are extremely grateful to you for providing suggestions to improve our manuscript quality. Also, as per your direction we have changed all the required modifications in the manuscript.

Reviewer #2: 

This manuscript has been greatly improved by the authors. In the process, however, several changes that the authors said they made are missing from the revised manuscript.

I also made some suggestions for language improvements to the revisions in the Discussion and Conclusions.

Author response: Thank you for the comments and we are extremely sorry and apologize for the missing items in the revised manuscript. This time we have carefully considered all your comments and corrected the revised manuscript.

Reviewer comment 1: Material and Methods

1. Sample size: State that 4 CgH days is considered the MCID between groups and include a citation.

Author response: Thank you for the comments and as per your suggestion the required changes have been made in the statistical analysis part. The MCID and SD are obtained from the previous pilot study, which was mentioned in the section and the study was not published. 

Reviewer comment 2: 2. Analysis: Baseline comparisons (Comment 11 from the first review):

The authors state that they removed the statistical comparisons of baseline variables between groups. This is not the case. Statements of no significance and/or p-values still appear in the following and these must be removed:

Abstract: Results: First sentence. - Removed

Results: Participants: Two sentences on demographics and clinical variables. - Removed 

(The new last sentence is supposed to replace the statistical comparisons with p-values). - Modified

Table 1: last column - Removed

Table 3: Baseline column - Removed

Author response: Thank you for the comments and as per your suggestions the required changes have been done in the manuscript.

Reviewer comment 3: 3. Analysis: Intention-to-Treat:

This has parts:

You added that all patients were included in the analysis even if data were missing.

Also add that participants were analyzed in the group to which they were randomized.

Author response: Thank you for the comments and as per your suggestions the required changes have been made in the manuscript.

Reviewer comment 4: Results 4. Partial eta-squared: (comment 16 from first review):

As per the response to reviewer comments, partial eta-squared has been removed from the text.

However, it remains to be removed from Table 2.

Author response: Thank you for the comments and as per your suggestions the required changes have been made in the table 2.

Reviewer comment 5: 5. Table 2: (Comment 17 from first review):

This has not been done as stated by the authors in the response to reviewer comments.

“Show score range for NDI (0-100 with 100 worst) and EQ-5D (0-100 with 100 best) in the second column.”

Author response: Thank you for the comments and as per your suggestions the required changes have been made in the table 2.

Reviewer comment 6: Discussion 6. 1st Paragraph:

Start the paragraph like this: “This was the first powered randomized trial to compare the effects of cervical and thoracic manipulation for patients with CgH. Cervical manipulation was found to be superior to thoracic manipulation and conventional PT (massage) for improving days with CgH, as well as headache and neck pain and disability, to 6 months.”

Make sure to check the grammar.

Briefly elaborate the contradiction with Borusiak et al. What did these authors find?

Author response: Thank you for the comments and as per your suggestions the required changes have been made in the 1st paragraph of discussion.

Reviewer comment 7: 7. 2nd Paragraph:

Start the paragraph like this: “The mechanism of action has yet to be determined. Manipulation of the cervical spine may promote afferent nerve fiber activity through stimulation of the cervical

joint receptors. It may improve…”

Author response: Thank you for the comments and as per your suggestions the required changes have been made in the 2nd paragraph of discussion.

Reviewer comment 8: 8. 3rd Paragraph: Start the paragraph like this: “Thoracic manipulation was also found to be more effective than conventional PT in improving both the primary and secondary outcomes.”

Author response: Thank you for the comments and as per your suggestions the required changes have been made in the 3rd paragraph of discussion.

Reviewer comment 9: Conclusion 9. Try this wording: “The current randomized controlled trial found that cervical spine manipulation was more effective in improving pain parameters (intensity, frequency and threshold), functional disability and quality of life in patients with cervicogenic headache than thoracic spine manipulation and conventional physiotherapy. Future studies are recommended to identify the biomechanical and biochemical mechanisms underlying the clinical and functional changes engendered by manipulation in the treatment of CgH.”

Author response: Thank you for the comments and as per your suggestions the required changes have been made in the conclusion.

---

## [Decision Letter · Decision Letter 2]

12 Feb 2024

PONE-D-23-12627R2Comparative effectiveness of cervical vs thoracic spinal-thrust manipulation for care of cervicogenic headache: a randomized controlled trial.

PLOS ONE

Dear Dr. Nambi,

Thank you for submitting your manuscript to PLOS ONE. After careful consideration, we feel that it has merit but does not fully meet PLOS ONE’s publication criteria as it currently stands. Therefore, we invite you to submit a revised version of the manuscript that addresses the points raised during the review process.

Dear authors, thank you for your replies. In the last review, I kindly asked you to indicate the page and lines of each change, but you submitted the article without this information. This complicates and delays the review process. Next time, I will consider this a reason to reject your article. In this review, in addition to highlighting the helped text, you need to report the page (and lines) of each of the changes.

Please follow the last adjustment of the article indicated by the peer-reviewer and me and submit it.

—Regarding minimum clinically important difference (MCID), please report the MCID for each of your outcomes and report it in the results. Outcomes that do not have an MCID established in the literature should be highlighted in the study limitations.

—Delete the "Sr. No." column in Tables 1 and 2.

—Table 1: Remove the p-value column.

—Table 2: This table remains problematic and unintuitive. I'll give an example: the p-value of 0.012** in the first row indicates that the groups have a significant difference, but the table describes three groups, and we know that the TSM and CPT groups have similar scores at baseline (TSM: 17.2 ± 1.9 vs. CPT 17.4 ± 1.7 [p-value?]), while CSM is different from them (16.8 ± 1.8). there are 3 independent comparisons (e.g., 1: TSM vs. CPT [p-value]; 2: TSM vs. CSM [p-value]; 3: CPT vs. CSM [p-value]), so it is necessary to report 3 p-values (one for each comparison). What is important for the RCT is not the overall variance, but the difference between the groups at different times (as in Table 3).

How do you fix this error?

First, delete the p-value column from Table 2 (leaving only the descriptive data); second, add a column to Table 3 and do the same analysis for the baseline time. I have made an example for you (available for download), go to this link and use this model and its headers+legends (Tables 1, 2, and 3):

< https://docs.google.com/document/d/1NdsUtg3x0BSAmpJ3kKI9rDYy_8VL6slb/edit?usp=sharing&ouid=104821689851272179944&rtpof=true&sd=true >

—Finally, if you need to adjust the results session wording because of the table changes, do so.

Kind regards,

André Pontes-Silva

Academic Editor

PLOS ONE

We look forward to receiving your revised manuscript.

Kind regards,

André Pontes-Silva

Academic Editor

PLOS ONE

Journal Requirements:

Reviewers' comments:

Reviewer's Responses to Questions

**Comments to the Author**

1. If the authors have adequately addressed your comments raised in a previous round of review and you feel that this manuscript is now acceptable for publication, you may indicate that here to bypass the “Comments to the Author” section, enter your conflict of interest statement in the “Confidential to Editor” section, and submit your "Accept" recommendation.

Reviewer #2: (No Response)

2. Is the manuscript technically sound, and do the data support the conclusions?

Reviewer #2: Yes

3. Has the statistical analysis been performed appropriately and rigorously? 

Reviewer #2: Yes

4. Have the authors made all data underlying the findings in their manuscript fully available?

Reviewer #2: Yes

5. Is the manuscript presented in an intelligible fashion and written in standard English?

Reviewer #2: Yes

6. Review Comments to the Author

Reviewer #2: 1. Not all the p-values for baseline comparisons have been removed. Please remove them:

Results: Participants: The p-value language has not been removed from the end of the paragraph "p > .05". Remove it.

Table 1: last column. The p-values column has not been removed. Remove it.

2. Table 3: At the bottom of the table, identify d: “d – Cohen’s d (effect size)”

7. PLOS authors have the option to publish the peer review history of their article (what does this mean?). If published, this will include your full peer review and any attached files.

Reviewer #2: No

---

## [Author Response · Author response to Decision Letter 2]

22 Feb 2024

Editor comments:

Editor comment: Dear authors, thank you for your replies. In the last review, I kindly asked you to indicate the page and lines of each change, but you submitted the article without this information. This complicates and delays the review process. Next time, I will consider this a reason to reject your article. In this review, in addition to highlighting the helped text, you need to report the page (and lines) of each of the changes.

Please follow the last adjustment of the article indicated by the peer-reviewer and me and submit it.

Author response: Dear Editor, we are extremely sorry and apologize for the above issue happened from our side and we know how much difficult it is to review the comments suggested by you and other reviewers. Therefore, in this review we have included the page number and line number for all the corrections we have made and thank you for giving this opportunity to resubmit our article. 

Editor comment 1: —Regarding minimum clinically important difference (MCID), please report the MCID for each of your outcomes and report it in the results. Outcomes that do not have an MCID established in the literature should be highlighted in the study limitations.

Author response: Thank you for the suggestion and as per your recommendation the MCID values of all the outcomes were added in the results section and highlighted with red color.

Page Number: 13 Line number: 27-31

Editor comment 2: —Delete the "Sr. No." column in Tables 1 and 2.

Author response: Thank you for the suggestion and as per your recommendation the "Sr. No." column in Tables 1 and 2 was removed.

Page Number: 20, 21 Line number: 1

Editor comment 3: —Table 1: Remove the p-value column.

Author response: Thank you for noticing the above mistake, and as per your suggestion the last column in the table 1 has been removed.

Page Number: 20 Line number: 1

Editor comment 4: —Table 2: This table remains problematic and unintuitive. I'll give an example: the p-value of 0.012** in the first row indicates that the groups have a significant difference, but the table describes three groups, and we know that the TSM and CPT groups have similar scores at baseline (TSM: 17.2 ± 1.9 vs. CPT 17.4 ± 1.7 [p-value?]), while CSM is different from them (16.8 ± 1.8). there are 3 independent comparisons (e.g., 1: TSM vs. CPT [p-value]; 2: TSM vs. CSM [p-value]; 3: CPT vs. CSM [p-value]), so it is necessary to report 3 p-values (one for each comparison). What is important for the RCT is not the overall variance, but the difference between the groups at different times (as in Table 3).

How do you fix this error? First, delete the p-value column from Table 2 (leaving only the descriptive data); second, add a column to Table 3 and do the same analysis for the baseline time. I have made an example for you (available for download), go to this link and use this model and its headers+legends (Tables 1, 2, and 3):

< https://docs.google.com/document/d/1NdsUtg3x0BSAmpJ3kKI9rDYy_8VL6slb/edit?usp=sharing&ouid=104821689851272179944&rtpof=true&sd=true >

Author response: Thank you for this important suggestion and as per your recommendation the p-values in the table 2 has been removed. 

The p values for each comparison (CSM vs TSM, CSM vs CPT and TSM vs CPT) is highlighted in the Table -3. 

As per the recommendation of second reviewer the column for the baseline comparison was removed from the Table 3 during the second review.

The header and footer of the Tables 1, 2, and 3 were modified as per your recommendation.

Page Number: 20-22 Line number: 1

Editor comment 5: —Finally, if you need to adjust the results session wording because of the table changes, do so.

Author response: Thank you for the suggestion and required changes have been done in the results section.

Page Number: 12 Line number: 27

Reviewer #2: 

Reviewer comment 1: Not all the p-values for baseline comparisons have been removed. Please remove them: Results: Participants: The p-value language has not been removed from the end of the paragraph" p > .05". Remove it.

Author response: Thank you for noticing the above mistake, and as per your suggestion the sentence has been removed.

Page Number: 12 Line number: 25

Reviewer comment 2: Table 1: last column. The p-values column has not been removed. Remove it.

Author response: Thank you for noticing the above mistake, and as per your suggestion the last column in the table 1 has been removed.

Page Number: 20 Line number: 1

Reviewer comment 3: Table 3: At the bottom of the table, identify d: “d – Cohen’s d (effect size)”

Author response:

Page Number: 22 Line number: 4

---

## [Editor Report · Decision Letter 3]

5 Mar 2024

Comparative effectiveness of cervical vs thoracic spinal-thrust manipulation for care of cervicogenic headache: a randomized controlled trial.

PONE-D-23-12627R3

Dear Dr. Nambi, we’re pleased to inform you that your manuscript has been judged scientifically suitable for publication and will be formally accepted for publication once it meets all outstanding technical requirements.

I hope that you will send new manuscripts to our journal.

Kind regards,

André Pontes-Silva

Academic Editor

PLOS ONE

---

## [Editor Report · Acceptance letter]

21 Mar 2024

PONE-D-23-12627R3 

PLOS ONE

Dear Dr. Nambi, 

I'm pleased to inform you that your manuscript has been deemed suitable for publication in PLOS ONE. Congratulations! Your manuscript is now being handed over to our production team.

Kind regards, 

on behalf of

Professor André Pontes-Silva 

Academic Editor

PLOS ONE